# Observation of the polaronic character of excitons in a two-dimensional semiconducting magnet CrI₃

Wencan Jin [1,6], Hyun Ho Kim[2,7], Zhipeng Ye[3], Gaihua Ye[3], Laura Rojas[3], Xiangpeng Luo[1], Bowen Yang[2], Fangzhou Yin[2], Jason Shih An Horng[1], Shangjie Tian [4], Yang Fu[4], Gongjun Xu[5], Hui Deng [1], Hechang Lei [4], Adam W. Tsen[2], Kai Sun [1], Rui He [3✉] & Liuyan Zhao [1✉]

Exciton dynamics can be strongly affected by lattice vibrations through electron-phonon coupling. This is rarely explored in two-dimensional magnetic semiconductors. Focusing on bilayer CrI₃, we first show the presence of strong electron-phonon coupling through temperature-dependent photoluminescence and absorption spectroscopy. We then report the observation of periodic broad modes up to the 8th order in Raman spectra, attributed to the polaronic character of excitons. We establish that this polaronic character is dominated by the coupling between the charge-transfer exciton at 1.96 eV and a longitudinal optical phonon at 120.6 cm$^{-1}$. We further show that the emergence of long-range magnetic order enhances the electron-phonon coupling strength by ~50% and that the transition from layered antiferromagnetic to ferromagnetic order tunes the spectral intensity of the periodic broad modes, suggesting a strong coupling among the lattice, charge and spin in two-dimensional CrI₃. Our study opens opportunities for tailoring light-matter interactions in two-dimensional magnetic semiconductors.

[1] Department of Physics, University of Michigan, 450 Church Street, Ann Arbor, MI 48109, USA. [2] Institute for Quantum Computing, Department of Chemistry, and Department of Physics and Astronomy, University of Waterloo, Waterloo, 200 University Ave W, Ontario N2L 3G1, Canada. [3] Department of Electrical and Computer Engineering, 910 Boston Avenue, Texas Tech University, Lubbock, TX 79409, USA. [4] Department of Physics and Beijing Key Laboratory of Opto-electronic Functional Materials & Micro-nano Devices, Renmin University of China, Beijing 100872, China. [5] Department of Statistics, University of Michigan, 1085 South University, Ann Arbor, MI 48109, USA. [6] Present address: Department of Physics, Auburn University, 380 Duncan Drive, Auburn, AL 36849, USA. [7] Present address: School of Materials Science and Engineering, Kumoh National Institute of Technology, Gumi, Gyeongbuk 39177, Korea. ✉email: rui.he@ttu.edu; lyzhao@umich.edu

The polaronic effect[1], which describes the strong coupling between charge and lattice vibrations, has a key role in a broad class of novel quantum phenomena ranging from colossal magnetoresistance[2] to anomalous photovoltaic effect[3]. In particular, the polaronic effect on excitons can profoundly modulate exciton dynamics upon photoexcitation and has been employed to describe intriguing optical and optoelectronic properties in materials such as hybrid organic-inorganic perovskite solar cells[4–6]. Compared with three-dimensional (3D) bulk systems, two-dimensional (2D) atomic crystals possess a couple of unique advantages in exploring the polaronic effect on exciton dynamics. First, the reduced dielectric screening in atomically thin samples enhances both the excitonic effect[7] and the electron–phonon (e-ph) coupling[8], which is expected to promote the polaronic effect of excitons. Second, unlike bulk materials in which the e-ph coupling is largely determined by intrinsic electronic and phonon band structures with limited tunability, 2D materials provide greater flexibility for engineering e-ph coupling through a number of approaches including carrier doping[9,10] and interfacial coupling[11–15], as well as dimensionality modulation[8], and therefore hold high promise for the future development of optoelectronic devices.

The realization of a long-range magnetic order in 2D semiconducting CrI₃ paves the way to engineer optical and optoelectronic properties of 2D magnetic semiconductors[16–23]. The large excitonic effect[24] from the localized molecular orbitals, of neither Wannier-type in 2D TMDCs nor Frenkel-type in ionic crystals, can be considered as the microscopic origin of the giant magneto-optical Kerr effect[25] and magnetic circular dichroism[26] signals in 2D CrI₃. Meanwhile, the strong e-ph coupling is suggested to cause the large Stokes shift, profound broadness, and skewed lineshape in the photoluminescence (PL) spectra of 2D CrI₃[26]. The coexistence of excitons and strong e-ph coupling in 2D CrI₃ naturally leads to open experimental questions of whether polaronic character emerges in the exciton dynamics and whether they are affected by the long-range magnetic order.

One fingerprint for the polaronic effect is the development of phonon-dressed electronic bands that appear as satellite bands in proximity to the original undressed one. Such features manifest as multiple equally spaced replica bands in angle-resolved photoemission spectroscopy (ARPES)[9,14,27–32] or as discrete absorption and emission lines in linear optical spectroscopy[33]. However,

such signatures of the polaronic effect have not been revealed so far in 2D CrI₃, as the sizable bandgap (~1.1 eV)[26] and extreme surface sensitivity[34] of CrI₃ make ARPES measurements challenging, whereas the potential inhomogeneous broadening could largely smear out individual lines for phonon-dressed satellite bands in linear optical spectroscopy.

In this work, we exploit temperature and magnetic field-dependent resonant micro-Raman spectroscopy, to show the direct observation of the polaronic character of excitons in bilayer CrI₃. The polaronic effect manifests in Raman spectra as a well-defined, periodic pattern of broad modes that is distinct from sharper phonon peaks. The profile of this periodic pattern and its temperature and magnetic field dependence further reveal essential information including the e-ph coupling strength and the tunability of polaronic effect by the magnetism in bilayer CrI₃. We mainly focus on bilayer CrI₃ because it features a single magnetic phase transition from the layered antiferromagnetic (AFM) to ferromagnetic (FM) order and briefly compare to the results on thicker CrI₃ flakes afterwards.

## Results

**Excitonic transitions and strong electron–phonon coupling.** We start by identifying excitonic transitions and e-ph coupling in bilayer CrI₃ using temperature-dependent PL and linear absorption spectroscopy. Bilayer CrI₃ was fully encapsulated between few-layer hexagonal BN (hBN) and placed on a sapphire substrate (for details, see "Methods"). Linear absorption spectroscopy measurements were then performed in a transmission geometry (see "Methods"). Figure 1 shows representative PL and absorbance spectra taken at 80 K, 40 K, and 10 K that correspond to well above, slightly below, and well below the magnetic critical temperature $T_C = 45$ K, respectively[25,26,34]. A single PL mode at 1.11 eV and three prominent absorbance peaks at 1.51 eV, 1.96 eV, and 2.68 eV (denoted as A, B, and C, respectively) are observed across the entire temperature range. These three energies are in good agreement with the ligand-field electronic transitions assigned by differential reflectance measurements on monolayer CrI₃[26] and bulk CrI₃[35,36] and have been later revealed to be bright exciton states through sophisticated first principle GW and Bethe-Salpeter equation calculations[24]. The large Stokes shift (~400 meV) between the PL and A exciton absorption peak

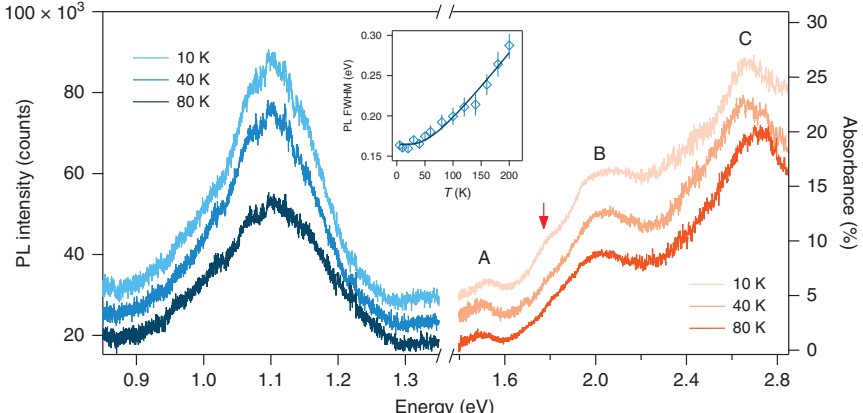

**Fig. 1 Exciton transitions in bilayer CrI₃.** Photoluminescence (PL) (left, blue) and absorption (right, orange) spectra of a bilayer CrI₃ encapsulated between two hBN flakes and placed on the sapphire substrate, at 10, 40, and 80 K. A, B, and C denote three main exciton transitions at 1.51, 1.96, and 2.68 eV, and the orange arrow marks a shoulder mode at 1.79 eV appearing only at low temperatures. Spectra at 10 K and 40 K are offset vertically for clarity. Inset shows the fitted full width at half maximum (FWHM) of PL spectra as a function of temperature, $\Gamma(T)$, (diamond symbols) and its fitting to the functional form $\Gamma(T) = \Gamma_0 + \dfrac{\gamma}{\exp\left(\frac{\hbar\omega_{LO}}{k_B T}\right) - 1}$ with the first and second terms for impurity-related inhomogeneous broadening and e-ph coupling-induced homogeneous broadening, respectively. Error bars indicate one standard error in fitting the FWHM of PL spectra.

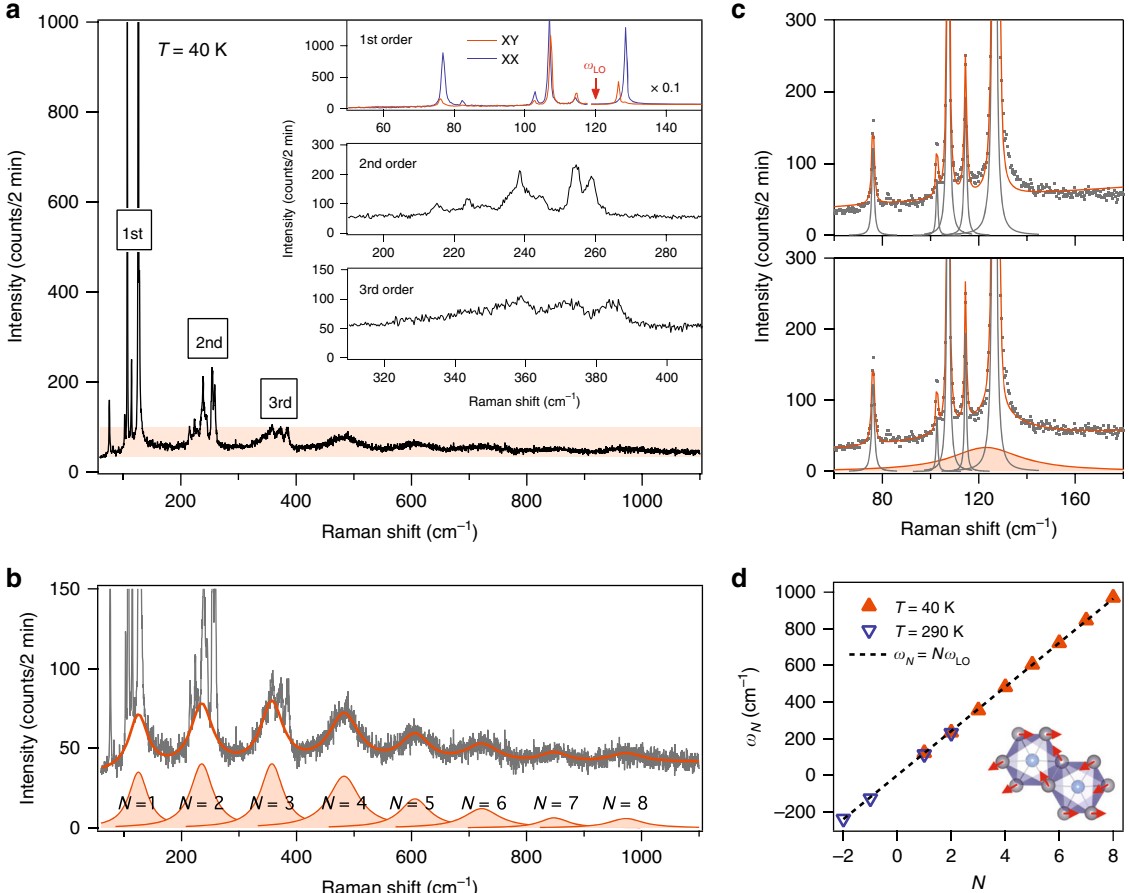

**Fig. 2 Polaronic character of exciton dynamics in bilayer CrI₃. a** Raman spectrum of bilayer CrI₃ acquired in the linearly crossed polarization channel at 40 K using a 633 nm laser. Insets show 1st-order single-phonon modes (linear parallel/crossed, i.e., XX/XY, channel in blue/red), 2nd-order two-phonon modes, and 3rd order three-phonon modes. The red arrow indicates the Raman-inactive longitudinal optical (LO) phonon frequency of importance, $\omega_{LO}$. The spectral intensities in the frequency range above 118 cm⁻¹ are scaled by a factor of 0.1. **b** Zoom-in of the orange shaded area in the spectrum in **a**. The solid orange line overlaid on the raw data is the fit to the periodic pattern in the Raman spectrum with a sum of eight Lorentzian profiles indexed from $N = 1$ to $N = 8$ and a constant background, i.e., $\sum_{N=1}^{8} \frac{A_N \left(\frac{\Gamma_N}{2}\right)^2}{(\omega - \omega_N)^2 + \left(\frac{\Gamma_N}{2}\right)^2} + C$. **c** Fit of the Raman spectrum over the 1st-order spectral range without (upper panel) and with (lower panel) the consideration of the $N = 1$ broad mode (shaded Lorentzian profile in the lower panel). **d** Plot of the fitted central frequency ($\omega_N$) of the Nth Lorentzian profile in data taken at 40 K (solid orange triangle) and 290 K (hollow blue triangle). Dashed line is a linear fit ($\omega_N = N\omega_{LO}$) to the plot that yields a slot of $\omega_{LO} = 120.6 \pm 0.9$ cm⁻¹. Inset shows the atomic displacement of the LO phonon mode.

is consistent with previous report[6] and indicates strong electron–phonon coupling in 2D CrI₃. Although the absorbance spectra show little temperature dependence except for the appearance of a weak shoulder at 1.79 eV at 10 K (orange arrow), the PL spectra are clearly temperature dependent. In particular, the temperature dependence of the PL full width at half maximum, $\Gamma(T)$, is well fitted by the model functional form, $\Gamma(T) = \Gamma_0 + \frac{\gamma}{\exp\left(\frac{\hbar\omega_{LO}}{k_B T}\right) - 1}$, with the first term for temperature-independent inhomogeneous broadening and the second term for homogeneous broadening from the exciton coupling with a longitudinal optical (LO) phonon at frequency $\omega_{LO}$. Taking $\omega_{LO} = 120.6$ cm⁻¹ found later on in Fig. 2, we obtain $\Gamma_0 = 163.9 \pm 2.7$ meV and $\gamma = 164.2 \pm 8.1$ meV, which suggests that the broadness of the exciton modes arises from both inhomogeneous broadening from disorders and homogeneous broadening from e-ph coupling. The large homogeneous broadening parameter ($\gamma$) indicates strong vibronic modes mixing in the PL spectra, which precludes the formation of well-resolved phonon sidebands[6].

**Polaronic character in Raman spectra**. We next proceed to perform resonant micro-Raman spectroscopy measurements with an incident wavelength of 633 nm matching the energy of the B exciton on an encapsulated bilayer CrI₃ flake placed on a SiO₂/Si substrate (see "Methods"). Figure 2a displays a representative Raman spectrum acquired in the crossed linear polarization channel at 40 K (slightly below $T_C = 45$ K). Note that this spectrum covers a much wider frequency range than earlier Raman studies on CrI₃[34,37–44]. The multiphonon scattering is visible up to the 3rd order, and their zoom-in Raman spectra are shown in the inset of Fig. 2a. The 1st-order single-phonon peaks appear in the relatively low frequency range of 50–150 cm⁻¹, and are assigned to be of either $A_g$ or $E_g$ symmetries under the $C_{3i}$ point group (see Supplementary Note 1), which is consistent with earlier work[34,37–44] and proves the high quality of our samples. The 2nd-order two-phonon and the 3rd-order three-phonon modes show up in slightly higher frequency ranges of 190–290 cm⁻¹ and 310–410 cm⁻¹, respectively, and show decreasing mode intensities at higher-order processes, same as typical multiphonon overtones under harmonic approximation[45] or cascade model[46]. In addition

to and distinct from these multiphonon features, we resolve a remarkable periodic modulation across a wide frequency range of 70–1100 cm$^{-1}$ in the low intensity part of the Raman spectrum (highlighted by the orange shaded area in Fig. 2a). This low intensity periodic pattern consists of clean, individual Lorentzian profiles and survives up to the 8th order (Fig. 2b), well beyond the highest order (3rd order) of multiphonon overtones, and each order of it spans for ~50 cm$^{-1}$ frequency range, much wider than the linewidth of any observed phonon modes (insets of Fig. 2a for phonons). Such a periodic pattern is also observed in the anti-Stoke's side at higher temperatures in bilayer CrI$_3$, for example, up to the 2nd order at 290 K (see Supplementary Note 2), which clearly supports its Raman origin instead of luminescence.

We fit this low intensity periodic pattern using a summation of Lorentzian profiles of the form $\sum_N \frac{A_N \left(\frac{\Gamma_N}{2}\right)^2}{(\omega - \omega_N)^2 + \left(\frac{\Gamma_N}{2}\right)^2} + C$ with central frequency $\omega_N$, linewidth $\Gamma_N$, and peak intensity $A_N$ of the $N$th period and a constant background $C$ (see fitting procedure in "Methods"). Among all eight orders ($N = 1, 2, \ldots, 8$.) in Fig. 2b, the presence of the 1st-order broad mode is deliberately validated in Fig. 2c that fitting with this 1st-order broad mode (orange shaded broad peak in the bottom panel) is visibly better than without it (top panel). This improved fitting by involving the 1st-order broad mode is further rigorously confirmed by the bootstrap method[47] (see Supplementary Note 3). Figure 2d shows a plot of the central frequency $\omega_N$ as a function of the order $N$ with data taken at 40 K ($N = 1, 2, \ldots, 8$) and 290 K ($N = -2, -1, \ldots, 3$), from which a linear regression fit gives a periodicity of 120.6 ± 0.9 cm$^{-1}$ and an interception of 0 ± 0.2 cm$^{-1}$. To the best of our knowledge, such a periodic pattern made of individual Lorentzian profiles previously has only been seen in multiphonon Raman spectra of Cd, Yb, and Eu monochalcogenides described by configuration-coordinate model[48–55]. However, the periodic pattern observed in bilayer CrI$_3$ here differs from these monochalcogenide multiphonon modes, as the broad linewidth of 1st-order mode contradicts with the sharp 1st-order forbidden LO phonon in Cd and Yb monochalcogenides[48–51] and the persistence (or even enhancement) of higher-order multiphonon below $T_C = 45$ K is in stark contrast to the disappearance of paramagnetic spin disorder-induced multiphonon below magnetic phase transitions in Eu monochalcogenides[52–55]. Because no known multiphonon model can capture all characteristics of our observed periodic pattern as well as the broad linewidths of each mode, we are inspired to consider the electronic origin. Indeed, strikingly similar features have been seen in polaron systems through the energy dispersion curves (EDCs) of ARPES[9,14,27–32] and linear absorption and PL spectroscopy[5,33,56]. In those cases, the periodic patterns in their energy spectra arise from the phonon-dressed electronic state replicas, or sometimes also referred as phonon-Floquet states[57], and the periodicity is given by the frequency of the coupled phonon. Owing to the high resemblance between the lineshapes of our Raman spectrum and those polaron energy spectra[5,14,27–32,56], we propose that this periodic pattern in Raman spectra of 2D CrI$_3$ stems from inelastic light scattering between the phonon-dressed electronic states caused by the polaronic character of B excitons in 2D CrI$_3$, whereby the B exciton at 1.96 eV, with the electron(hole) in the weakly dispersive conduction (highly dispersive valence) band of Cr 3$d$ (I 5$p$) orbital character[24,58], couples strongly to a phonon at 120.6 cm$^{-1}$[59]. It is worth noting that a recent theoretical work predicts magnetic polaronic states in 2D CrI$_3$ because of charge-magnetism coupling[60], whereas our work suggests polaronic exciton states due to charge-lattice coupling.

We then proceed to identify the source and the character of the phonon at 120.6 cm$^{-1}$. We first rule out the possibility of this phonon arising from either the hBN encapsulation layers or the SiO$_2$/Si substrate, as a similar periodic pattern in the Raman spectrum is also observed in bare bulk CrI$_3$ crystals (see Supplementary Note 4). Compared with the calculated phonon band dispersion of monolayer CrI$_3$[61], we then propose the LO phonon calculated to be at ~115 cm$^{-1}$ as a promising candidate, whose slight energy difference from the experimental value of 120.6 cm$^{-1}$ could result from the omission of e-ph coupling in calculations. This LO phonon mode belongs to the parity-odd $E_u$ symmetry of the $C_{3i}$ point group, and its atomic displacement field transforms like an in-plane electronic field ($E_x$, $E_y$) (see inset of Fig. 2d)[61]. Its odd parity makes it Raman-inactive and absent in the 1st-order phonon spectra (Fig. 2a inset, top panel), whereas its polar displacement field allows for its strong coupling to electrons/holes and prompts the polaronic character of the charge-transfer B exciton (see Supplementary Note 5 for measurements with additional laser wavelengths). In addition, this LO phonon band is nearly dispersionless and has a large density of states, further increasing its potential for coupling with the B exciton in 2D CrI$_3$.

**Temperature dependence of the polaronic effect**. Given the coexistence of a 2D long-range ferromagnetic order and polaronic effect of excitons below $T_C = 45$ K in bilayer CrI$_3$, it is natural to explore the interplay between the two. For this, we have performed careful temperature-dependent Raman spectroscopy measurements and fitted the periodic pattern in every spectrum with a sum of Lorentzian profiles. Figure 3a displays the periodic pattern in Raman spectra taken at 70 K and 10 K, well above and below $T_C$, respectively. Comparing these spectra, not only do more high-order replica bands become visible at lower temperatures (i.e., from $N = 6$ at 70 K to $N = 8$ at 10 K), but also the spectral weight shifts toward the higher-order bands (i.e., from $N = 1$ at 70 K for the strongest mode to between $N = 3$ and 4 at 10 K in Fig. 3b). The appearance of higher-order modes at lower temperatures possibly results from a combination of the narrow exciton linewidth (~50 cm$^{-1}$) and the dispersionless nature of coupled LO phonon. More importantly, the spectral weight distribution ($A_N$ vs. $N$) quantifies the e-ph coupling strength, and its spectral shift across $T_C$ confirms the interplay between the polaronic effect and the magnetic order in bilayer CrI$_3$. Theoretically, the polaron system consisting of dispersionless LO phonons and charges is one of the few exactly solvable models in many-body physics[62], and the calculated polaron spectra can be well-described by a Poisson distribution function, $A_N = A_0 \frac{e^{-\alpha} \alpha^N}{N!}$[63,64], where $A_0$ is the peak intensity of the original electronic band, $A_N$ is the peak intensity for the $N$th replica band with $N$ phonon(s) dressed, and $\alpha$ is a constant related to the e-ph coupling in 3D (i.e., $\alpha_{3D}$) that can be scaled by a factor of $3\pi/4$ for 2D (i.e., $\alpha_{2D}$)[65]. By fitting the extracted Lorentzian peak intensity profile at every temperature to the Poisson distribution function (see fits of 10 K and 70 K data in Fig. 3b), we achieve a comparable fitting quality to that for ARPES EDCs in polaron systems[9,31] at every temperature and eventually arrive at the temperature dependence of $\alpha_{2D}$, which remains nearly constant until the system is cooled to $T_C$ and then increases by almost 50% at the lowest available temperature 10 K of our setup (Fig. 3c). In addition to the anomalous enhancement of $\alpha_{2D}$ across $T_C$, the value of $\alpha_{2D} = 1.5$ at 10 K is the highest among known 2D polaron systems including graphene/BN heterostructures ($\alpha_{2D} = 0.9$)[14] and bare SrTiO$_3$ surfaces ($\alpha_{2D} = 1.1$)[29].

**Magnetic field dependence of the polaronic effect**. It has been shown that bilayer CrI$_3$ transitions from a layered AFM to FM with increasing out-of-plane magnetic field ($B_\perp$) above the critical value $B_C$ of 0.7 T[17,21,25,26]. We then finally explore the evolution

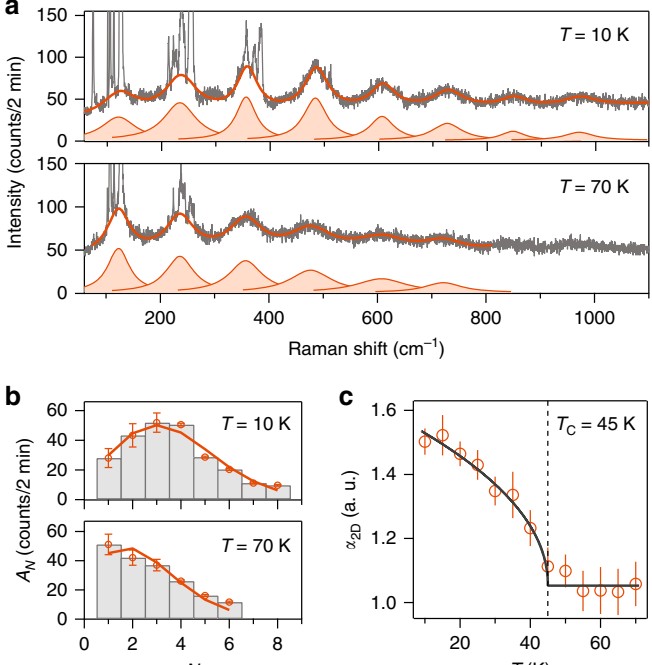

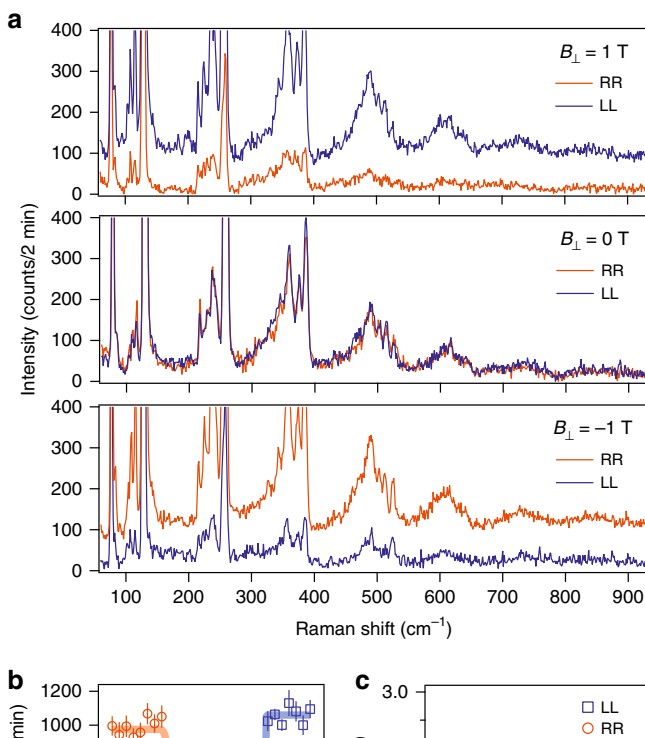

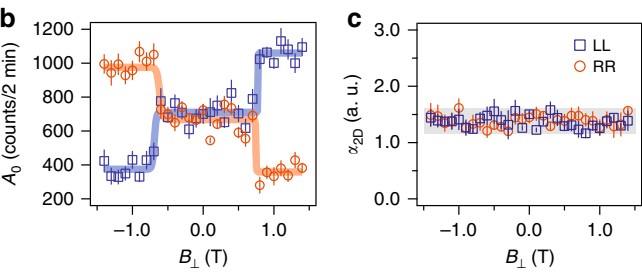

**Fig. 3 Enhanced e-ph coupling across the magnetic onset $T_C$ in bilayer CrI$_3$. a** Raman spectra of bilayer CrI$_3$ acquired at 10 K and 70 K, respectively. Solid orange lines are fits to the raw Raman spectra, using a sum of $N$ Lorentzian profiles and a constant background, i.e.,

$$\sum_N \frac{A_N \left(\frac{\Gamma_N}{2}\right)^2}{(\omega - \omega_N)^2 + \left(\frac{\Gamma_N}{2}\right)^2} + C.$$ **b** Histogram plot of the fitted Lorentzian mode intensity ($A_N$) as a function of order index ($N$) at 10 K and 70 K. Solid curves are fits of the peak intensity profiles to the Poisson distribution function, $A_N = A_0 \frac{e^{-\alpha} \alpha^N}{N!}$. Error bars represent one standard deviation that convolutes the standard errors from the fittings to the local individual and global multiple Lorentzian profiles. **c** Plot of two-dimensional electron–phonon coupling constant ($\alpha_{2D}$) as a function of temperature. The dashed vertical line marks the magnetic onset $T_C = 45$ K, and the solid line is the fit to the functional form $\alpha_{2D}(T) = \begin{cases} A\sqrt{T_C - T} + B; & T < T_C \\ B; & T > T_C \end{cases}$. Error bars in **c** represent one standard error in the Poisson fitting.

**Fig. 4 Evolution of polaronic effect across the magnetic phase transition from the layered AFM to FM in bilayer CrI$_3$. a** Raman spectra of bilayer CrI$_3$ acquired at 10 K in the circularly parallel polarization channels, RR and LL, where RR (LL) stands for that both incident and scattered polarizations are selected to be right-handed (left-handed) circularly polarized, with an applied out-of-plane magnetic field ($B_\perp$) of 1 T (top), 0 T (middle), and −1 T (bottom), respectively. **b, c** Plots of the Poisson fit amplitude $A_0$ (**b**) and electron–phonon coupling strength $\alpha_{2D}$ (**c**) as a function of the applied $B_\perp$ in RR (orange data points) and LL (blue) channels. Solid lines are step (orange and blue in **b**) and linear (gray in **c**) function fits to the magnetic field dependence of $A_0$ and $\alpha_{2D}$, respectively. Error bars are defined as one standard error of the fitting parameters.

of the polaronic effect across this magnetic phase transition by performing magnetic field-dependent Raman spectroscopy measurements. Here, we choose circularly polarized light to perform magnetic field-dependent measurements in order to eliminate any Faraday effect from the optical components that are situated in close proximity to the strong magnetic field. Figure 4a shows Raman spectra taken at $B_\perp = 0$ T and ±1 T, below and above $B_C$, respectively, in both RR and LL channels, where RR(LL) stands for the polarization channel selecting the right-handed (left-handed) circular polarization for both incident and scattered light (see Supplementary Note 6). At 0 T, the spectra are identical in the RR and LL channels, consistent with zero net magnetization in the layered AFM state for bilayer CrI$_3$ at $|B_\perp| < B_C$. At ±1 T, the spectra in the RR and LL channels show opposite relative intensities under opposite magnetic field directions, owing to the fact that the net magnetization in the FM state for bilayer CrI$_3$ at $|B_\perp| > B_C$ breaks the equivalence between the RR and LL channels. To better quantify the magnetic field dependence of the spectra, we measured Raman spectra in the RR and LL channels at $B_\perp$ from −1.4 T to 1.4 T every 0.1 T. We fit the spectrum at every magnetic field to extract $A_N$ first and then $A_0$ and $\alpha_{2D}$.

Figure 4b shows that $A_0$ has abrupt changes at $B_\perp = \pm 0.7$ T in both RR and LL channels, consistent with the first order magnetic phase transition at $B_C$. Furthermore, the magnetic field dependence of $A_0$ shows an opposite trend in the RR channel from that in the LL channel, whereas the sum of $A_0$ from both channels remain nearly constant to the varying magnetic field. This observation can be understood by that, under a time-reversal operation, the RR channel transforms into the LL channel and the direction of the net magnetization at $|B_\perp| > B_C$ flips, resulting in that the Raman spectrum in the RR channel at $B_\perp > 0.7$ T is equivalent to the spectrum in the LL channel at $B_\perp < -0.7$ T. Figure 4c shows that $\alpha_{2D}$ is magnetic field independent, suggesting that the interlayer magnetic order barely affects the e-ph coupling strength and that the in-plane long-range magnetic order is responsible for the strong enhancement of e-ph coupling at $T_C$. This finding corroborates with the fact that the 120.6 cm$^{-1}$ phonon has in-plane atomic displacement.

## Discussions

Our further Raman spectroscopy studies on tri-layer, four-layer, and five-layer $CrI_3$ show qualitatively same findings as those in bilayer $CrI_3$ (see Supplementary Note 7) and again echoes with the in-plane nature of the 120.6 $cm^{-1}$ $E_u$ phonon and the intralayer charge-transfer B exciton. Our data and analysis reveal the phonon-dressed electronic states and suggest the polaronic character of excitons in 2D $CrI_3$, which arises from the strong coupling between the lattice and charge degrees of freedom and is dramatically modified by the spin degree of freedom of $CrI_3$. The exceptionally high number of phonon-dressed electronic state replicas (up to $N = 8$) further suggests 2D $CrI_3$ as an outstanding platform to explore nontrivial phases out of phonon-Floquet engineering, whereas the significant coupling to the spin degree of freedom adds an extra flavor whose impact on the phonon-Floquet states has not been studied. For example, one can imagine creating topological states through the band inversion between the phonon-dressed replicas of $CrI_3$ and the electronic state of a material in close proximity.

## Methods

**Sample fabrication.** $CrI_3$ single crystals were grown by the chemical vapor transport method, as detailed in ref. [40]. Bilayer $CrI_3$ samples were exfoliated in a nitrogen-filled glove box. Using a polymer-stamping transfer technique inside the glove box, bilayer and few-layer $CrI_3$ flakes were sandwiched between two few-layer hBN flakes and transferred onto $SiO_2$/Si substrates and sapphire substrates for Raman spectroscopy and PL/linear absorption spectroscopy measurements, respectively.

**Linear absorption spectroscopy.** A bilayer $CrI_3$ sample on a sapphire substrate was mounted in a closed-cycle cryostat for the temperature-dependent absorption spectroscopy measurements. A broadband tungsten lamp was focused onto the sample via a 50× long working distance objective. The transmitted light was collected by another objective and coupled to a spectrometer with a spectral resolution of 0.2 nm. The absorption spectra were determined by $1 - \frac{I_{sample}(\lambda)}{I_{substrate}(\lambda)}$, where $I_{sample}(\lambda)$ and $I_{substrate}(\lambda)$ were the transmitted intensity through the combination of sample and substrate and through the bare substrate, respectively.

**PL spectroscopy.** PL spectra were acquired from the same bilayer $CrI_3$ sample where we carried out linear absorption measurements. The sample was excited by a linearly polarized 633 nm laser focused to a ~2 μm spot. A power of 30 μW was used, which corresponds to a similar fluence reported in the literature [26] (10 μW over a 1 μm-diameter spot). Transmitted right-handed circularly polarized PL signal was dispersed by a 600 grooves/mm, 750 nm blaze grating, and detected by an InGaAs camera.

**Raman spectroscopy.** Resonant micro-Raman spectroscopy measurements were carried out using a 633 nm excitation laser for the data in the main text and 473 nm, 532 nm, and 785 nm excitation lasers for data in Supplementary Note 5. The incident beam was focused by a 40× objective down to ~3 μm in diameter at the sample site, and the power was kept at ~ 80 μW. The scattered light was collected by the objective in a backscattering geometry, then dispersed by a Horiba LabRAM HR Evolution Raman spectrometer, and finally detected by a thermoelectric cooled CCD camera. A closed-cycle helium cryostat is interfaced with the micro-Raman system for the temperature-dependent measurements. All thermal cycles were performed at a base pressure lower than $7 \times 10^{-7}$ mbar. In addition, a cryogen-free magnet is integrated with the low temperature cryostat for the magnetic field-dependent measurements. In this experiment, the magnetic field was applied along the out-of-plane direction and covered a range of $-1.4$ to $+1.4$ Tesla. In order to avoid the Faraday rotation of linearly polarized light as it transmits through the objective under the stray magnetic field, we used circularly polarized light to perform the magnetic field-dependent Raman measurements.

**Fitting procedure.** For every sample, we have taken temperature and magnetic field-dependent Raman spectra on the hBN/$SiO_2$/Si substrate with the same experimental conditions as that on the $CrI_3$ flakes. The Raman spectra from the substrate, an extremely gradual background with a Si phonon peak at ~525 $cm^{-1}$, shows no dependence on temperature (over the range of 10–70 K) and magnetic field (0–2.2 T). To fit the periodic oscillations in Raman spectra of $CrI_3$ flakes, we follow the procedure described below. (I) we fit the Si phonon peak at ~525 $cm^{-1}$ in both spectra taken on the $CrI_3$ thin flake and the bare substrate to extract the Si peak intensity, $I_{Si}^{sample}$ and $I_{Si}^{substrate}$. (II) we multiply the background spectrum by a factor of $\frac{I_{Si}^{sample}}{I_{Si}^{substrate}}$, which is ~1, and then subtract off the factored background from the raw Raman spectrum of sample. This process leads to the pure Raman signal for

$CrI_3$ whose baselines are nearly identical over the temperature range of interest (10–70 K). (III) we fit the sharp $CrI_3$ phonon peaks with Lorentzian functions and subtract their fitted functions from the background free Raman spectrum from step (II). This leads to a clean spectrum with only periodic broad modes for a global fitting. (IV) we fit the clean spectrum from step (III) with a sum of multiple Lorentzian functions, $\sum_N \frac{A_N \left(\frac{\Gamma_N}{2}\right)^2}{(\omega - \omega_N)^2 + \left(\frac{\Gamma_N}{2}\right)^2} + C$. For the neatness of the data presentation in Figs. 2 and 3, we only show the fitted line from step IV in the plots.

## Data availability

The data sets generated and/or analyzed during the current study are available from the corresponding author on reasonable request.

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

## Acknowledgements

We thank X. Xu, M. Kira, R. Merlin, X. Qian, and H. Wang for useful discussions. L. Zhao acknowledges support by NSF CAREER grant no. DMR-1749774. R. He acknowledges support by NSF CAREER grant no. DMR-1760668 and NSF MRI grant no. DMR-1337207. K. Sun acknowledges support through NSF grant no. NSF-EFMA-1741618. A. W. Tsen acknowledges support from the US Army Research Office (W911NF-19-10267), Ontario Early Researcher Award (ER17-13-199), and the National Science and Engineering Research Council of Canada (RGPIN-2017-03815). This research was undertaken, thanks in part to funding from the Canada First Research Excellence Fund. H. Lei acknowledges support by the National Key R&D Program of China (grant no. 2016YFA0300504), the National Natural Science Foundation of China (no. 11574394, 11774423, and 11822412), the Fundamental Research Funds for the Central Universities, and the Research Funds of Renmin University of China (15XNLQ07, 18XNLG14, and 19XNLG17). H. Deng and J. Horng acknowledge support by the Army Research Office under Awards W911NF-17-1-0312.

## Author contributions

W.J., R.H., and L.Z. conceived this project and designed the experiment; S.T., Y.F., and H.L. synthesized and characterized the bulk CrI₃ single crystals; H.H.K., B.Y., F.Y., and A.W.T. fabricated and characterized the few-layer samples; Z.Y., G.Y., and L.R. performed the Raman measurements under the guidance of R.H. and L.Z.; J.H., W.J., and H.D. performed the linear absorption and PL spectroscopy measurements; W.J., X.L., R.H., and L.Z. analyzed the data with discussions with K.S.; G.X. provided statistical modeling in data analysis; W.J., X.L., R.H., and L.Z. wrote the paper and all authors participated in the discussions of the results.

## Competing interests

The authors declare no competing interests.
