## [Peer Review File · Nature Communications]

REVIEWER COMMENTS

Reviewer #1 (Remarks to the Author):

The manuscript demonstrated the evidence of polarons related to B exciton and LO phonon in CrI₃ via Raman spectroscopy. The author tried to confirm their assignments based on the necessary of broad band in the fitting, the Raman intensity following Poisson distribution function and the absence of Raman-inactive first-order LO phonon mode. Further temperature and magnetic-field dependence of the periodic oscillations are demonstrated to highlight the strong coupling between exciton, phonon and long-range magnetic order. As CrI₃ is an important 2D magnet and have attracted much attentions in the past three years, this work would be timely and of potential interest in the field of magnetic 2D materials. The following issues should be clarified:

1. In the manuscript, the author tried to confirm that the broad band would be necessary for the fitting based on bootstrap method. Intuitively, the weak intensity and broadening of multiphonon modes in principle lead to the overlap of different high-order Raman modes and thus result in the so-called broad periodic oscillations, especially for the higher-order (>4th order) multiphonon modes. Furthermore, the high-order Raman band is given by the convolution of various multiphonon processes across the whole Brillouin zone and therefore, might not be fitted by a sum of Lorentzian curves, as present in Ref (Doi: 10.1038/ncomms14670). From the section 3 of the SI, we can also see that only small improvement after including a weak broad band. It seems that the multiphonon modes can fit well for the experimental data. Fundamentally, how to directly confirm that these periodic oscillations are exactly related to the so-called polarons from a physical view? It would be very important and basic for the polaron studies via Raman spectroscopy. I know that the author has highlighted that the Raman intensity of the polaron modes follow a Poisson distribution function in the manuscript. However, the Raman intensity of these broad bands would be very sensitive to the fitting process. For example, for the fitting of 2nd and 3rd period only considered small contributions from the sharp multiphonon Raman modes (Fig.2b and Fig.3a), it would result in an increasing intensity of the broad bands. That might be the reason why the author would obtain higher intensity for A2 and A3. In addition, no matter from the polarization behaviors, the temperature dependence or magnetic-field responses, the periodic oscillations and multiphonon modes exhibit similar tendency. So, it is very confusing that how to tell apart the signals of polarons from multi-phonon scattering based on the experimental data.
2. The authors mentioned that "the phonon mode involved in the formation of the polarons should be parity odd". Why it should be parity odd? Can the author give a simple demonstration about this, since the absence of odd-parity peak at 120.6 cm⁻¹ is one of the evidences for the polaron modes? The cited ref. 58 of the manuscript just demonstrated the electrons in lattice fields and the effective mass of electron in crystals, but not any discussions about the polaron, phonon.
3. In page 4/17, the author heighted that "This observation is further supported by a recent theoretical work whereby polaronic effects of excitons in CrI₃ is predicted to be present²⁹". However, in ref. 29, the magnetic polaron is predicted to be formed in electron-doped bilayer CrI₃ but not in pristine bilayer CrI₃, which is indeed related to electron but not exciton. Thus, this theoretical work seems not to provide any support for the exciton-polaron discussed in this manuscript.
4. From the data shown in Fig.S2, there is a broad background. Is it universal for all the Raman spectra? If yes, how to determine the baseline and perform subtraction? Different subtraction would lead to varied fitted intensity of periodic oscillations.
6. From Fig.3b, there is apparent deviation for the fitting of Raman intensity of polaron modes to the Poisson distribution function. Can the author give insights into the reason for the deviation?
7. The title of this manuscript is "Evidence of the polaronic character of excitons in a two-

dimensional semiconducting magnet CrI₃"; however, all the shown data are from bilayer CrI₃. As the ferromagnetic properties in CrI₃ significantly depends on the layer number, it may play an important role in the coupling of exciton, phonon and long-range magnetic order. Thus, it would be great to give insight into the layer-number dependence of the periodic oscillation. Is it possible to observe the so-called periodic oscillations in monolayer CrI₃?

8. As pointed out in the manuscript and Fig.S6, the observed periodic oscillations are related to the resonant Raman effect. However, the Raman spectra shown in Fig. S6 is under the excitation far from the B exciton (633 nm). It is difficult to distinguish the periodic oscillations even under 532 nm. I suggest the author to perform the Raman measurements with laser energy slightly closer to the B exciton (e.g., 647 nm). Due to the broad linewidth of absorption related to B exciton transition, the periodic oscillations should be emergent in a large excitation energy range, though the intensity would drastically reduce. The resonance profile would provide solid evidence for the claim of resonant Raman scattering.

Reviewer #2 (Remarks to the Author):

In this work, the authors report the observation of periodic peaks in the Raman spectrum of CrI₃ and interpret them as being due to exciton-polarons. The experimental data are of high quality and the work seems to have been carried out with care. This work would attract interest from the researchers working on 2-dimensional magnetic materials. This manuscript can be published in Nature Communications after the authors address the following issues.

1. The large Stokes shift and the broad linewidth of the photoluminescence (PL) peak are not properly explained. From the temperature dependence, the authors explain that the broadening comes from a combination of inhomogeneous and homogeneous broadening. However, broadening parameters of ~ 160 meV are huge, especially at low temperatures, and so a proper explanation should be offered. The large Stokes shift of ~ 400 meV should also be explained. This has nothing to do with the large exciton binding energy because the first absorption peak at 1.51 eV is already the lowest energy exciton. Since the authors do not present the absorption spectra in the near-infrared range, I assume that the absorption at ~ 1.1 eV is negligible. Then what is the origin of this PL? In traditional semiconductors, such a PL signal far below the lowest absorption edge would be classified as being due to midgap states due to defects. In order to argue against such an explanation, the authors should present a convincing model to explain the origin of this PL.

2. The authors observed the periodic signals only with the 633 nm (1.98 eV) excitation which is resonant with the B exciton state, but no such signal was observed with the 785 nm and 473 nm excitations which are (approximately) resonant with the A and C exciton states, respectively. How can one argue that only the B exciton forms the exciton-polaron quasiparticles?

3. The authors explain that the phonon involved in forming the exciton-polarons is the Raman-inactive Eu phonon at 120.6 cm⁻¹. Because the whole interpretation in this work hinges on the presence of this phonon, it is paramount to confirm that such a phonon exists. The authors cite a theoretical work that predicts an Eu phonon at 115 cm⁻¹, but given the importance of this phonon, independent experimental verification is a must. Under resonant excitation, normally forbidden modes tend to appear in the Raman spectrum due to the Fröhlich interaction. I wonder why there is no hint of a mode at 120 cm⁻¹ in the Raman spectrum. If the mode cannot be seen in Raman scattering, infrared absorption can be measured relatively easily since the sample is already on a transparent substrate. Without some experimental evidence for a phonon mode at ~ 120 cm⁻¹, the entire paper is on very shaky ground.

4. The shift of the spectral weight to higher orders at 10 K is not explained. A reasonable explanation for this observation would offer strong support for the authors' interpretation.

5. I wonder why the authors chose not to cite their own recent work [PRX 10, 011075 (2020)] and another Raman work [Nat. Nano. 15, 212 (2020)] on the same material.

Point-by-point response

Reviewer #1 (Remarks to the Author):

The manuscript demonstrated the evidence of polarons related to B exciton and LO phonon in CrI₃ via Raman spectroscopy. The author tried to confirm their assignments based on the necessary of broad band in the fitting, the Raman intensity following Poisson distribution function and the absence of Raman-inactive first-order LO phonon mode. Further temperature and magnetic-field dependence of the periodic oscillations are demonstrated to highlight the strong coupling between exciton, phonon and long-range magnetic order. As CrI₃ is an important 2D magnet and have attracted much attentions in the past three years, this work would be timely and of potential interest in the field of magnetic 2D materials. The following issues should be clarified:

We thank Reviewer #1 for taking his/her time reading and evaluating our work. We appreciate his/her positive comments on the interest of our work, as well as the constructive questions below in improving our manuscript. Below please see our point-by-point response to the issues/questions raised by him/her.

1. In the manuscript, the author tried to confirm that the broad band would be necessary for the fitting based on bootstrap method. Intuitively, the weak intensity and broadening of multiphonon modes in principle lead to the overlap of different high-order Raman modes and thus result in the so-called broad periodic oscillations, especially for the higher-order (>4th order) multiphonon modes. Furthermore, the high-order Raman band is given by the convolution of various multiphonon processes across the whole Brillouin zone and therefore, might not be fitted by a sum of Lorentzian curves, as present in Ref (Doi: 10.1038/ncomms14670). From the section 3 of the SI, we can also see that only small improvement after including a weak broad band. It seems that the multiphonon modes can fit well for the experimental data. Fundamentally, how to directly confirm that these periodic oscillations are exactly related to the so-called polarons from a physical view? It would be very important and basic for the polaron studies via Raman spectroscopy. I know that the author has highlighted that the Raman intensity of the polaron modes follow a Poisson distribution function in the manuscript. However, the Raman intensity of these broad bands would be very sensitive to the fitting process. For example, for the fitting of 2nd and 3rd period only considered small contributions from the sharp multiphonon Raman modes (Fig.2b and Fig.3a), it would result in an increasing intensity of the broad bands. That might be the reason why the author would obtain higher intensity for A2 and A3. In addition, no matter from the polarization behaviors, the temperature dependence or magnetic-field responses, the periodic oscillations and multiphonon modes exhibit similar tendency. So, it is very confusing that how to tell apart the signals of polarons from multi-phonon scattering based on the experimental data.

We thank Reviewer #1 for his/her recognition of this work being important for polaron studies via Raman spectroscopy. We note that, to our knowledge, this is perhaps the first work of using Raman spectroscopy to capture polaronic characteristics in 2D magnets, and therefore we very much appreciate the questions from Reviewer #1 to solidify our statements and interpretations.

Below we first address the key question from Reviewer #1 “are the periodic oscillations different from multiphonon modes?” and then answer a few closely related and more detailed questions.

Key question: Are the periodic oscillations different from multiphonon modes?

Response:

1. To address this question, we first review all three different types of the multiphonon modes reported in the literature and summarize their signatures in Raman spectra as follows:

(a) Harmonic approximation model multiphonon

This is the most common mechanism for multiphonon in Raman spectra. A very sharp 1st-order phonon mode can be observed, while the scattering cross-sections of higher-order modes become exceedingly small due to the iterations of single phonon process. In addition, due to the requirement of momentum conservation, every higher-order multiphonon feature of this type comes from different phonons across the entire Brillouin zone with compensated momenta (*i.e.*, a total momentum of participated phonons is zero) and thus often have complex structures, rather than a single, clean Lorentzian profile. A good example of this is B. R. Carvalho *et al.*, Nat. Comm. **8**, 14670 (2017).

(b) Cascade model multiphonon

This model (R. M. Martin and C. M. Varma Phys. Rev. Lett. **26**, 1241 (1971)) was proposed to explain multiphonon modes of shifted-LO phonons in some monochalcogenides (*e.g.*, CdS, ZnTe, *etc* Refs. 1-5 in Phys. Rev. Lett. **26**, 1241 (1971)) and attribute their origin to emission of multiple energy quanta of an individual phonon in the excited electronic state. A very sharp 1st-order phonon mode is expected, while the peak intensities of higher-order multiphonon modes decay exponentially and their linewidths increase significantly. Compared with (a), a single phonon mode of zero momentum participates in this process, and its first- and higher-order phonon modes all exhibit clean, single Lorentzian profiles.

(c) Configuration-coordinate model multiphonon

Configuration-coordinate model is based on the fact that the equilibrium positions of nuclei change between the ground and excited electronic states and therefore has strong electron-phonon coupling. This model has often been used to explain multiphonon in a set of materials with relatively localized electronic states and quantify the multiphonon intensity *v.s.* its order index by a Poisson distribution. We summarized them into three groups based on the origin of the 1st-order phonon.

c1. multiphonon of the forbidden-LO phonon.

One classic example of this is YbS (R. Merlin *et al.*, Phys. Rev. B **17**, 4951 (1978)), where the forbidden-LO phonon is accessed by the Fröhlich interactions between the LO phonon at a finite momentum and the localized electronic states (the exciton states in YbS). Here, a sharp 1st-order LO phonon mode with a shifted frequency from the zero momentum LO-phonon is observed, and higher-order multiphonon modes have increasing linewidths. Also noted is that both 1st- and higher-order phonon modes in this case are only present in the fully symmetric channels (*i.e.*, parallel channel).

c2. multiphonon of the lattice-disorder assisted phonon.

This is often observed in alloys or chemically-doped materials where atomic defects and/or dopants are commonly present. The atomic defects/dopants break lattice translational symmetry to access otherwise forbidden phonon modes, but often lead to relatively broad phonon linewidths.

c3. multiphonon of the spin-disorder assisted phonon.

This is typically present in paramagnetic phases of magnetic materials (e.g., Eu-monochalcogenides in Phys. Rev. B **9**, 984 (1974)) where the spin disorders break translational symmetry of the underlying electronic states and thus make otherwise forbidden phonon modes Raman active. Here, even the 1st-order mode is very broad due to spin disorder, and no further linewidth broadening in higher-order modes is observable. Moreover, such multiphonon modes vanish in the magnetic ordered phase (below T_M and under external magnetic fields) because of the restoration of translational symmetry.

2. We then compare the periodic pattern in our study with the three multiphonon mechanisms listed above, and we find none of them can explain all the characteristics of the periodic oscillations in our data. We discuss the contradiction between our observations and the three multiphonon models as follows:

(a) We can rule out the possibility of the observed periodic oscillations being multiphonon from the harmonic approximation and cascade models based on two facts:

First, the broadness of the 1st-order mode in our periodic pattern contradicts with the sharp 1st-order phonon mode in both models. Second, the mode intensity v.s. mode index in our data shows a Poisson-like distribution (for instance, with the peak centered at the 3rd order at 40 K), instead of a monotonic decay trend dictated by both models.

(b) We also note clear discrepancies between our data and the configuration-coordinate model multiphonon of each case:

Forbidden LO-phonon: The absence of a sharp 120 cm^{-1} 1st-order phonon mode and the presence of the periodic pattern in both parallel and crossed channels in our data distinguish our data from the multiphonon of forbidden LO-phonon.

Lattice disorder-induced phonon: The very sharp 1st-order phonons (linewidths as narrow as $\sim 0.8\text{ cm}^{-1}$) in our data confirm the high quality of our samples and simultaneous the absence of disorder effect in our data.

Spin disorder-induced phonon: The persistence, or even enhancement, of the periodic pattern below $T_C = 45\text{ K}$ differentiates our results from the multiphonon of spin disorder-induced phonon.

From the comparison above, we can see that the periodic pattern observed in our data are distinct from the known examples of multiphonon in the literature. We therefore move to a larger literature pool to look for suitable explanations. Indeed, we found that ARPES measurements on polaronic systems exhibit periodic electronic bands separated by multiples of phonon energies (Refs. 2, 17, 27–32 in the main text). Please see more details in the answer to the related question Q2 below.

Related questions

Q1: How to reliably fit the periodic oscillations when (multi)phonon modes are present?

When the strong sharp phonon modes are present, it requires great care to fit the data and we have tried our best to do so. In fact, the stronger the phonon modes are, the more challenging the fit for the weak broad mode in the periodic oscillations is. This makes the 1st order is the hardest, and the

2nd order follows and then the 3rd order goes, when the sharp (multi)phonon modes are progressively weaker in higher orders.

- In the main text and SM, we have chosen the 1st order, the hardest, to illustrate our careful procedure: (a) fit the raw data with only sharp modes (Lorentzian functions) first to get decent estimate of the sharp modes parameters; (b) fit the raw data with both sharp modes and a broad mode (Lorentzian functions), with the initial input for the sharp modes parameters from (a); (c) perform bootstrap test to compare whether (a) or (b) is a true model for the raw data. As Reviewer #1 has pointed out, the raw-eye inspection for the difference between the two panels in Figure 2(c) of the main text is not impressive. We would like to point out that this “smallness” is comparing the misfit of ~ 20 -30 counts (the difference between the fitted line to the raw data) in method (a) with respect to the sharp peaks of ~ 3700 counts for the strongest one, making the improvement of method (b) seemingly unimpressive. However, comparing to the broad mode height of ~ 30 counts, the improvement brought by method (b) is truly important. The bootstrap test further quantifies the significance of this improvement that the chance for the raw data having no broad mode is as low as less than 5%.
- For the 2nd and 3rd orders, we have followed steps (a) and (b) above and show the presence of broad modes for both orders. For the sharp multiphonon modes in the 2nd and 3rd orders, we used a sum of Lorentzian functions to mimic their lineshapes, although the physical meaning of these Lorentzian profiles is not as defined as that in the 1st order. It serves the practical goal here to get the parameters for the broad mode beneath these sharp modes. We note that a more thorough understanding of the sharp multiphonon in 2nd and 3rd order is definitely an interesting and important question, but it is beyond the scope of this manuscript.
- For the 4th and higher orders, there are no sharp modes, and the raw data is well fit by single broad Lorentzian modes.
- We remove the sharp phonon modes from the raw spectra based on the fitting above and then perform a global fit of the periodic pattern.

Q2: How to directly confirm that the periodic oscillations are exactly related to the so-called polarons from a physical view?

We propose that the periodic pattern originates from electronic Raman scattering off the periodic phonon-dressed electronic states whose energy separations are equal to multiples (integer N) of the involved LO-phonon energy ($N \times \omega_{LO}$). This explanation is inspired from the ARPES results of other polaronic systems such as SrTiO₃ (Nature Comm. **6**, 8585 (2015), Nature Materials **15**, 835(2016)) (Figure R1a) and MoS₂ (Nature Materials **17**, 676 (2018)).

Adopting the same picture, we draw a simplified cartoon for the case of 2D CrI₃ in Figure R1b where the N periodic phonon-dressed bands are labeled. In our experiment, the incident light creates electron-hole pairs across a charge transfer transition between I $5p$ and Cr $3d$ orbitals, then the electron is scattered from the original undressed Cr $3d$ orbital state to the N^{th} phonon-dressed state, and finally the recombination of electron-hole pairs emits light with an energy difference of

$N\omega_{LO}$ from the incident light. We would like to point out the consistency between our data and the picture.

Fig. R1. **a.** an ARPES example showing phonon-dressed electronic states whose energies are separated by integer multiples of the LO-phonon frequency. From top to bottom: raw data, first derivative of the raw data, and a vertical energy line cut at momentum zero (Nature Materials, **15**, 835 (2016)). **b.** simplified cartoon illustration of the process in which electron scattered between the phonon-dressed electronic states in 2D CrI₃ (labeled in dashed blue).

- The broadness of each mode in the periodic pattern is consistent with the electronic origin of these phonon-dressed electronic states whose linewidths are dominated by the much broader electronic state linewidths.
- The atomic displacements of the 120.6 cm^{-1} phonon branch involve the in-plane vibrations between the Cr and I atoms, which is expected to directly modulate the charge transfer transition between the Cr $3d$ and I $5p$ orbitals (B exciton resonance). Furthermore, this 120.6 cm^{-1} phonon is of E_u symmetry, transforms as in-plane polar vector (x, y), and therefore creates internal electronic field (E_x, E_y) that can efficiently couple to the electronic degree of freedom (such as B exciton).
- The extracted mode intensity *v.s.* the order index N matches reasonably well with the expected Poisson distribution as expected for the phonon-dressed states (Nature Comm. **6**, 8585 (2015), Nature Materials **15**, 835(2016), and Nature Materials **17**, 676 (2018)) for a similar reason as the configuration-coordinate model.

Q3: How to compare the periodic oscillations to the complex multiphonon lineshape in Nat. Comm. 8, 14670 (2017)?

Reviewer #1 pointed out that the higher Raman band can arise from the convolution of various multiphonon processes across the entire Brillouin zone, which can create nontrivial Raman lineshapes that are not accountable by a simple sum of Lorentzian functions (Nat. Comm. 8, 14670 (2017)). We totally agree with the richness of such a multiphonon process and the complexity of the associated Raman spectrum lineshape, as nicely demonstrated in Nat. Comm. 8, 14670 (2017). We believe that this knowledge is very important in understanding the sharp multiphonon modes in the 2nd and 3rd orders in our data, and we would like to pursue in future studies.

In contrast to the complex lineshape of multiphonons which includes asymmetry and bumpiness of the spectra (Nat. Comm. 8, 14670 (2017)), individual broad modes in the periodic oscillations are simply single Lorentzian profiles, which is most apparent when the sharp multiphonon modes are gone in orders higher than the 4th. Such a clean single Lorentzian profile for the N^{th} broad mode also suggests a single scattering channel for it, rather than a complex convolution that involves various channels.

To address all these questions above, we have made the following revision to the manuscript:

- Modify the title to be “Observation of the polaronic character of excitons ...”
- Revise the discussions on the 1st-order phonons and its 2nd-and 3rd-order overtones to show their resemblance to the harmonic approximation and/or cascade model multiphonon (line 88 – 92).
- Add the comparison between the periodic pattern in this manuscript and configuration-coordinate model multiphonon in literature (line 110 – 119).
- Add the description of intuitive physical picture for such a periodic pattern in 2D CrI₃ (line 124 – 128).
- Add and revise detailed description of “Fitting procedure” section in Methods.

2. The authors mentioned that "the phonon mode involved in the formation of the polarons should be parity odd". Why it should be parity odd? Can the author give a simple demonstration about this, since the absence of odd-parity peak at 120.6 cm⁻¹ is one of the evidences for the polaron modes? The cited ref. 58 of the manuscript just demonstrated the electrons in lattice fields and the effective mass of electron in crystals, but not any discussions about the polaron, phonon.

We thank Reviewer #1 for these questions about the phonon mode. It should have odd parity (in a centrosymmetric crystal) because the parity-odd phonon can induce dynamic polarizations in the lattice and consequently lead to much stronger coupling with electrons/holes than parity-even modes. Based on this guidance, together with the absence of 120.6 cm⁻¹ phonon in the 1st-order phonon Raman spectra, we assign the 120.6 cm⁻¹ phonon mode to be an E_u mode whose displacement field can indeed produce an in-plane polar moment that transforms like (P_x, P_y) and induces an dynamic internal in-plane electric field to couple to electrons/holes.

We cited Ref. 58 in the old version (H. Fröhlich. Advances in Physics 3, 325(1954)) because it is the foundational paper that formulate the interactions between electrons and phonon-induced polarization for the first time. This Fröhlich model, including Fröhlich Hamiltonian and electron-

phonon coupling strength α , has been extensively used in literature, such as Refs. 2, 17, 27–32 cited in our manuscript.

To deliver our message more clearly, we have revised our manuscript to

- Include in Figure 2d the atomic displacement cartoon for this 120.6 cm^{-1} phonon mode and revised the Figure caption.
- Add and revise in-depth discussions about this 120.6 cm^{-1} phonon mode and its coupling to B excitons in 2D CrI_3 (line 137 – 144).

3. In page 4/17, the author highlighted that "This observation is further supported by a recent theoretical work whereby polaronic effects of excitons in CrI_3 is predicted to be present²⁹". However, in ref. 29, the magnetic polaron is predicted to be formed in electron-doped bilayer CrI_3 but not in pristine bilayer CrI_3 , which is indeed related to electron but not exciton. Thus, this theoretical work seems not to provide any support for the exciton-polaron discussed in this manuscript.

We appreciate Reviewer #1 for his/her sharp point on this issue, and we apologize for the inaccuracy in expressing our intention. We agree that the microscopic origins of magnetic polaron and exciton polaron are indeed different, although both of them describe electrons/holes "bound" to their local environment. Their microscopic difference is that the local environment is lattice vibration for exciton polaron (or traditional polaron) and spin exchange interactions for magnetic polaron. Our intention of citing Ref. 29 in the old version (Ref. 60 in the revised version) was to show the possibility of polaron formation in CrI_3 , and we now realize that we should have clarified the difference of microscopic origin between the two types of polarons.

To improve the accuracy of our presentation, we have revised the description when citing Ref. [60] (Ref. 29 in the last version) and made proper comparisons to our result (line 128 – 130).

4. From the data shown in Fig.S2, there is a broad background. Is it universal for all the Raman spectra? If yes, how to determine the baseline and perform subtraction? Different subtraction would lead to varied fitted intensity of periodic oscillations.

We are aware of the presence of the broad background, which is greater at higher temperature (such as room temperature). The spectrum in Figure S2 was taken at 290 K for the sake of detecting anti-Stokes signal to prove that the periodic oscillations are Raman signal instead of photoluminescence. But, over the temperature range of interest in this manuscript (10 K – 70 K) where we perform serious temperature dependence analysis across the magnetic phase transition at $T_C = 45$ K, the broad background is nearly identical, as we can see that the baselines for the Raman spectra at 10 K and 70 K overlap as shown in Figure R2.

Because of the broad, gradual nature of the background, it is approximated by a linear line with a very small slope over the Raman shift frequency range of 50 – 1000 cm^{-1} , the range that we focus in the manuscript. This background does not create any complication in fitting the periodic oscillations. Because of nearly identical background over 10 – 70 K, this background does not intervene with the temperature dependence of the periodic oscillations over 10 – 70 K.

Fig. R2. Raman spectra of 2L CrI₃ acquired at 10 K (blue) and 70 K (orange), respectively, to show their nearly identical background.

To convey this information, we have clarified that the broad background for 2D CrI₃ Raman spectra over 10 – 70 K are nearly identical in the Method section.

6. From Fig.3b, there is apparent deviation for the fitting of Raman intensity of polaron modes to the Poisson distribution function. Can the author give insights into the reason for the deviation?

We also noticed that the Poisson distribution function slightly under/overfit some data points in Fig. 3b, and such fitting errors have been faithfully included as error bars of the Poisson fitting parameter α at individual temperatures in Fig. 3c. Despite that these fitting errors are a few to ten percent of fitted α at every temperature, we can clearly see the order-parameter-like onset of α at the magnetic critical temperature T_C .

For the potential reason for these errors, we attribute to three factors to our best knowledge now. First, the signal level for the polaron feature is indeed very small, making its detection more likely to be influenced by any random noise/error. Second, the fitting procedure for extracting individual broad mode intensity for the periodic oscillation is nontrivial, because of the presence of strong phonon modes sitting on top of our interested broad modes in the periodic oscillations (as detailed above in question 1). Third, Poisson distribution itself is a more restrictive model whose mean equals to its variance, as oppose to other distributions with independent mean and variance. Therefore, it is more likely to see larger fitting errors in Poisson distributed data sets. We would like to point out that in the ARPES measurements of polaronic systems, the fitting of their energy dispersion curve (EDC) to the Poisson model (*e.g.*, Fig. 2g-j in Nature Materials **15**, 835(2016)) is of similar quality as our Raman data here.

To implement the answer above in our manuscript, we have revised to:

- Add the fitting error bars of each mode intensity in Figure 3b and update its figure caption.
- Include a comparison between the quality of our Poisson fitting of Raman spectra and the Poisson fitting of ARPES EDC in literature (line 162 – 164).

7. The title of this manuscript is "Evidence of the polaronic character of excitons in a two-dimensional semiconducting magnet CrI₃"; however, all the shown data are from bilayer CrI₃. As the ferromagnetic properties in CrI₃ significantly depends on the layer number, it may play an important role in the coupling of exciton, phonon and long-range magnetic order. Thus, it would be great to give insight into the layer-number dependence of the periodic oscillation. Is it possible to observe the so-called periodic oscillations in monolayer CrI₃?

We thank Reviewer #1 for bringing up this point. Indeed, monolayer (1L) CrI₃ is Ising ferromagnetic whose spin orientation can be flipped by a smaller than 0.15 T out-of-plane magnetic field (B. Huang *et al.*, Nature **546**, 270-273 (2017)); bilayer (2L) CrI₃ is a layered-antiferromagnet (AFM) which undergoes a layered-AFM to FM transition at a critical field of 0.65 T (B. Huang *et al.*, Nature **546**, 270-273 (2017)); and thicker CrI₃ films are also layered-AFM, but go through two critical magnetic fields, 0.7 T and 1.6 T corresponding to the spin-flip for the surface and interior layers respectively, till arriving at the final layered-FM state (D. R. Klein *et al.*, Science **360**, 1218-1222 (2018) and T. Song *et al.*, Science **360**, 1214-1218 (2018)).

In this manuscript main text, we have focused on bilayer CrI₃ for two reasons. First, the layered-AFM to FM phase transition only happens in 2L or thicker CrI₃. Second, we have studied 3L, 4L, and 5L CrI₃, and they all show qualitatively similar results as 2L CrI₃.

- In Figure R3, we show the temperature dependence of 3L, 4L, and 5L CrI₃, for which we have performed the same analysis as we did for 2L CrI₃ (shown in Fig. 3 of the main text). Spectral weight shift to higher orders below T_C is observed for all three thicknesses, same as what 2L CrI₃ behaves. This makes sense because the E_u phonon at $\sim 120.6 \text{ cm}^{-1}$ that couples strongly to the B exciton primarily has in-plane atomic displacement.
- In Figure R4, we show the magnetic field dependence of 3L, 4L, and 5L CrI₃, and we have performed the same analysis as we did for 2L CrI₃ (shown in Fig. 4 of the main text). The data successfully captures the magnetic phase transition at 1.6 T for all three thicknesses, qualitatively similar to that the data for 2L CrI₃ detects the transition into the fully spin polarized FM state. At the same time, we also comment that the data in Figure R4 does not pick up the magnetic phase transition at 0.7 T for all three thicknesses. This could be because that the intermediate magnetic phase between 0.7 T and 1.6 T has multiple degenerate states (D. R. Klein *et al.*, Science **360**, 1218-1222 (2018) and T. Song *et al.*, Science **360**, 1214-1218 (2018)) and therefore could have domains inside the probe beam, reducing the net signal change.

We have also measured the polaron spectrum in monolayer CrI₃ and found the signal level of the periodic oscillations in monolayer is quite weak (perhaps due to smaller sample volume and less ideal sample quality for monolayer CrI₃). Nonetheless, the presence of the polaron feature in monolayer and multilayer CrI₃ is consistent with the proposed in-plane E_u phonon mode that couples strongly to the B exciton in 2D CrI₃.

Fig. R3. Temperature dependent Raman spectra taken on **a.** 3L, **b.** 4L, and **c.** 5L CrI₃ with the same analysis as Fig. 3 in the main text.

Fig. R4. a. Magnetic field dependent Raman spectra taken on 3L, 4L, and 5L CrI₃ in the RR polarization channel with the same analysis as Fig. 4 in the main text. **b.** Magnetic field dependence of the fitted A_0 , the overall intensity of the Poisson profile.

To include this additional information about layer-number dependence, we have revised manuscript to

- Include a supplementary section 7 for the data of 3-5L CrI₃.
- Explain the motivation of focusing on 2L CrI₃ (line 55 – 57)
- Compare the 2L results to those of 3-5L (line 193 – 196).

8. As pointed out in the manuscript and Fig.S6, the observed periodic oscillations are related to the resonant Raman effect. However, the Raman spectra shown in Fig. S6 is under the excitation far from the B exciton (633 nm). It is difficult to distinguish the periodic oscillations even under 532 nm. I suggest the author to perform the Raman measurements with laser energy slightly closer to the B exciton (e.g., 647 nm). Due to the broad linewidth of absorption related to B exciton transition, the periodic oscillations should be emergent in a large excitation energy range, though the intensity would drastically reduce. The resonance profile would provide solid evidence for the claim of resonant Raman scattering.

We thank Review #1 for his/her suggestion. Indeed, it is a great idea to perform systematic wavelength dependent measurements, in particular, focusing on the excitation energy around the B exciton, to explore how the polaron spectra evolve as a function of excitation energy. Unfortunately, we don't have a tunable CW laser to perform such an exciting experiment, and the

four lasers used in Fig. S6 are all we have now. We would look for collaborations with other Raman groups who have this research capability.

We would like to clarify the reason why we call 633 nm excitation as resonant Raman: 633 nm laser matches the charge transfer transition in CrI₃ which has been demonstrated and widely accepted in literature (*e.g.*, Refs. 6, 8, 19, 20, 21, and 25). Here, we inherit this knowledge and terminology from literature, in addition to our independent confirmation by absorption spectra in Fig. 1 of the main text.

To clarify our intension of using these four laser lines for experiments, we have added further discussions on the Raman spectra taken with these four wavelengths in Supplementary Section 5.

In summary, we thank Reviewer #1 for his/her constructive questions in improving our manuscript.

Reviewer #2 (Remarks to the Author):

In this work, the authors report the observation of periodic peaks in the Raman spectrum of CrI₃ and interpret them as being due to exciton-polarons. The experimental data are of high quality and the work seems to have been carried out with care. This work would attract interest from the researchers working on 2-dimensional magnetic materials. This manuscript can be published in Nature Communications after the authors address the following issues.

We thank Reviewer #2 for his/her positive comments on our work and appreciate his/her potential recommendation of publication in Nature Communications. Below we address the issues/questions from Reviewer #2.

1. The large Stokes shift and the broad linewidth of the photoluminescence (PL) peak are not properly explained. From the temperature dependence, the authors explain that the broadening comes from a combination of inhomogeneous and homogeneous broadening. However, broadening parameters of ~ 160 meV are huge, especially at low temperatures, and so a proper explanation should be offered. The large Stokes shift of ~ 400 meV should also be explained. This has nothing to do with the large exciton binding energy because the first absorption peak at 1.51 eV is already the lowest energy exciton. Since the authors do not present the absorption spectra in the near-infrared range, I assume that the absorption at ~ 1.1 eV is negligible. Then what is the origin of this PL? In traditional semiconductors, such a PL signal far below the lowest absorption edge would be classified as being due to midgap states due to defects. In order to argue against such an explanation, the authors should present a convincing model to explain the origin of this PL.

Reviewer #2 is totally correct that the lowest absorption peak is at 1.5 eV and has a large Stokes shift of ~ 400 meV from the PL peak at 1.1 eV. Both this large Stokes shift and the broad linewidth of PL of 2D CrI₃ were also reported in Nature Physics **14**, 277 (2018), where differential reflection measurement was performed to capture the optical transitions (equivalently, absorption peaks).

We have cited this work as Ref. 6 in our manuscript and provide a summary of their interpretations below.

- Both the PL at 1.11 eV and the differential reflection peak at 1.51 eV originate from the ligand-field 4A_2 to 4T_2 transition ($d-d$ transition).
- The ~ 400 meV Stokes shift between 1.11 eV PL mode and 1.51 eV differential reflectance peak is due to the Franck-Condon principle and strong electron-lattice coupling.
- Meanwhile, the broad PL linewidth is a result of $d-d$ luminescence broadened by vibronic modes, in which the strong vibronic mixing precludes the formation of well-resolved phonon sidebands in the spectra. The phonon mode found here is at 194 cm^{-1} , whose direct experimental detection and physical properties (*e.g.*, symmetry) remain open.
- The possibility of PL from midgap defect states have been ruled out based on two observations: (i) the PL intensity of monolayer CrI_3 is linearly proportional to the excitation power, in contrast to the case for defect-induced PL whose intensity typically saturates at high excitation power; (ii) the PL of CrI_3 shows helicity that depends on the layered magnetic order, in contrast to the case for defect-induced PL which is not expected to show magnetic-order-dependent helicity.

In addition to the results in Nature Physics **14**, 277 (2018), we provided in this manuscript systematic temperature dependence of PL linewidth ($\Gamma(T)$) to extract the temperature-independent broadening (Γ_0) and the temperature-dependent phonon-coupled homogeneous broadening ($\frac{\gamma}{\exp(\frac{\hbar\omega_{\text{LO}}}{k_B T}) - 1}$), *i.e.*, $\Gamma(T) = \Gamma_0 + \frac{\gamma}{\exp(\frac{\hbar\omega_{\text{LO}}}{k_B T}) - 1}$ with $\omega_{\text{LO}} = 120.6\text{ cm}^{-1}$ from our periodic oscillation data. Our result here further supports the strong electron-phonon coupling in CrI_3 and motivates our systematic Raman studies.

To include the answer to this question in the manuscript, we have revised the main text to include discussions on the large Stokes shift and broad linewidth (line 69 – 70 and 78 – 80) and cited Nature Physics 14, 277 (2018) for their interpretations.

2. The authors observed the periodic signals only with the 633 nm (1.98 eV) excitation which is resonant with the B exciton state, but no such signal was observed with the 785 nm and 473 nm excitations which are (approximately) resonant with the A and C exciton states, respectively. How can one argue that only the B exciton forms the exciton-polaron quasiparticles?

We thank Reviewer #2 for this insightful question. From what we know, we attribute the observation of exciton-polaron at 633 nm (B exciton) but not at 785 nm and 473 nm excitations to two factors, a scientific one and a technical one.

- Scientifically, since the LO phonon involves in-plane atomic displacements between Cr and I atoms, making its coupling to the I 5p to Cr 3d charge transfer transition more efficient than any onsite transitions. Exciton A corresponds to $d-d$ transition between Cr 3d orbitals, and therefore, it is expected that the LO phonon-A exciton coupling is weak. In contrast, B exciton corresponds to the I 5p to Cr 3d (E_g manifold) charge transfer resonance, and therefore couples to the LO

phonon efficiently. Finally, C exciton is thought to be I 5p to Cr 3d (t_{2g} manifold), which in principle, should couple to the LO phonon as well.

- Technically, 633 nm matches better with the B exciton than the other two wavelengths to A and C excitons. As we can see from Figure R5 where the gradual background is removed from the absorption spectra in Fig. 1a in the main text, 785 nm only meets the upper tail of A exciton resonance and 473 nm is a bit on the lower energy side of the C exciton resonance, whereas 633 nm is nearly at the B exciton resonance. This is consistent with the much weaker phonon signals in the Raman spectra taken with 785 nm and 473 nm excitations than that with 633 nm excitation.

Fig. R5. Absorption spectroscopy with the gradual background removed to highlight the A, B, and C exciton modes and to compare with the four laser wavelengths used in this study.

As a result of the combination of scientific and technical factors, we only see phonon-dressed electronic states at 633 nm which matches the B exciton resonance. However, it does not mean that C exciton does not form exciton-polaron with this LO phonon at 120 cm^{-1} . Further Raman studies with tunable CW lasers to precisely match C exciton resonances are needed to clarify if their exciton-polaron states are present.

To implement these answers in our manuscript, we have added in Supplementary Section 5 the comparison the four laser wavelengths to the background-free absorption spectroscopy and further discussions on the Raman spectra taken at these four wavelengths.

3. The authors explain that the phonon involved in forming the exciton-polarons is the Raman-inactive Eu phonon at 120.6 cm^{-1} . Because the whole interpretation in this work hinges on the presence of this phonon, it is paramount to confirm that such a phonon exists. The authors cite a theoretical work that predicts an Eu phonon at 115 cm^{-1} , but given the importance of this phonon, independent experimental verification is a must. Under resonant excitation, normally forbidden modes tend to appear in the Raman spectrum due to the Fröhlich interaction. I wonder why there is no hint of a mode at 120 cm^{-1} in the Raman spectrum. If the mode cannot be seen in Raman scattering, infrared absorption can be measured relatively easily since the sample is already on a transparent substrate. Without some experimental evidence for a phonon mode at $\sim 120\text{ cm}^{-1}$, the entire paper is on very shaky ground.

We agree with Reviewer #2 that it would be best if we have direct experimental evidence of the E_u phonon mode at 120.6 cm^{-1} , and therefore we have tried our best to follow the two suggestions from him/her.

- Any hint of the 120.6 cm^{-1} mode in Raman spectrum because of the Fröhlich interaction and the resonant excitation?

It has been shown in previous studies that the forbidden LO phonon may become Raman-visible from the intra-band matrix element of the Fröhlich interactions and that the Raman efficiency is given by

$$S(\text{LO}, \omega) \sim \omega_{\text{LO}} \left(\frac{1}{\epsilon_-} - \frac{1}{\epsilon_+} \right) \left| \frac{1}{m_e} - \frac{1}{m_h} \right|^2 \left| \vec{e}_i \cdot \frac{d^2 \tilde{\epsilon}}{d\omega^2} \cdot \vec{e}_s \right|^2$$

where ω_{LO} is the LO phonon frequency, $\tilde{\epsilon}$ is the dielectric tensor, $\epsilon_{-/ +}$ is the real part of the dielectric function below/above ω_{LO} , and $m_{e/h}$ is the effective mass of electron/hole (Phys. Rev. B, **45**, 3037 (1992)). From this equation, we can see

- 1) The Raman efficiency critically depends on the resonance condition to maximize $\frac{d^2 \tilde{\epsilon}}{d\omega^2}$ and the choice of resonant band to maximize $\frac{1}{m_e} - \frac{1}{m_h}$.
- 2) The symmetry of the Fröhlich interaction induced forbidden LO-phonon is the same as the dielectric tensor $\tilde{\epsilon}$, which is of fully symmetric A_g irrep.

So in order to observe the forbidden LO-phonon in Raman spectra, one needs to fine tune the excitation energy to locate the maximum curvature in $\epsilon_{xx}(\omega)$ (*i.e.*, $= \epsilon_{yy}(\omega)$ given the in-plane structure of CrI_3) and the greatest $\frac{1}{m_e} - \frac{1}{m_h}$.

Unfortunately, we did not observe an A_g Raman mode at 120.6 cm^{-1} at all four available excitations (785 nm, 633nm, 532nm, and 473nm). The potential reasons could be:

- 1) While 633nm excitation matches best the charge transfer resonance that provides a large (but not necessarily the largest) $\frac{1}{m_e} - \frac{1}{m_h}$, it may not be fine-tuned enough to meet the maximum $\frac{d^2 \epsilon_{xx}}{d\omega^2}$.
 - 2) Given the broadness of B exciton, the curvature of $\epsilon_{xx}(\omega)$ may be rather small (seen from the absorption spectroscopy), and therefore even the maximum $\frac{d^2 \epsilon_{xx}}{d\omega^2}$ itself may not be large enough.
- Can infrared absorption spectroscopy be used to detect this mode at 120.6 cm^{-1} ?
In general, infrared absorption spectroscopy has been a challenging experiment for mechanically exfoliated 2D flakes, because the infrared wavelength is typically much larger than the lateral dimension of the 2D flakes. Successful infrared spectroscopy on 2D materials have been carried out either on CVD grown wafer-scale 2D films (*e.g.*, ACS Nano 5, 9854, (2011)) or on purposely selected large-area exfoliated flakes (*e.g.*, Nature Communications 8, 14071 (2017)). For the latter case, special designs of optical experiments are still needed to improve the signal-to-noise ratio.

Here, to observe the infrared absorption peak for the 120.6 cm^{-1} (15.0 meV) phonon mode, the light source should have a wavelength as long as $\sim 80 \mu\text{m}$ (15.5 meV), whose diffraction limited focused beam diameter is $\sim 80 \mu\text{m}$. Therefore, the CrI_3 flake suitable for the experiment should have a lateral dimension comparable to the diffraction limit of $\sim 80 \mu\text{m}$ wavelength light. From our experience of exfoliating CrI_3 (statistics of nearly 100 flakes), the typical lateral size of thin CrI_3 flakes is in order of a few to ten μm , far smaller than the desired $\sim 80 \mu\text{m}$.

Despite the direct experimental evidence of 120.6 cm^{-1} E_u phonons, we would like to highlight a few points to support our assignment of this phonon for the polaron formation.

- 1) Phonon calculations are typically very reliable, provided that the atomic structure is accurate. Here, the CrI_3 atomic structure is relatively simple and has been experimentally established by diffraction measurements (M. A. McGuire *et al.*, Chem. Mater. **2**, 612-620 (2015)). And the reliability of the calculated phonons is supported by the consistency of these (7) Raman active phonon modes between calculations and Raman experiments. Therefore, the presence of an E_u phonon at a frequency of around 115 cm^{-1} is plausible.
- 2) The calculated E_u phonon frequency differing slightly from the experimental one could result from a couple of factors. (i) the calculations were done for monolayer CrI_3 without the consideration of interlayer coupling for thicker flakes. (ii) the calculations did not account for the electron-phonon coupling which could renormalize the phonon frequency slightly. Therefore, the $\sim 5.6 \text{ cm}^{-1}$ difference between the calculation and experiment is in fact quite reasonable.
- 3) E_u phonon creates a dynamics in-plane polarization, which is consistent with the facts that the observed periodic pattern has no clear layer-number dependence (Supplementary Section 7) and that the electron-phonon coupling does not dependence on the interlayer magnetism (Fig. 4c of main text).

To include the further information in the manuscript, we have revised the manuscript to

- Add further discussions on this 120.6 cm^{-1} E_u phonon in the main text (line 137 – 144).
 - Add layer-number dependent data in Supplementary Section 7 that supports the assignment of this E_u phonon with in-plane displacement field.
4. The shift of the spectral weight to higher orders at 10 K is not explained. A reasonable explanation for this observation would offer strong support for the authors' interpretation.

The intensity A_N of individual modes in the periodic oscillations vs. the order index N follows the Poisson distribution $A_N = A_0 \frac{e^{-\alpha} \alpha^N}{N!}$, where A_0 represents the overall intensity and α determines the spectral weight distribution. Physically, α stands for the electron-phonon coupling strength.

The spectral weight shift as a function of temperature is therefore quantified as the temperature dependence of α as shown in Fig. 3c of main text. We observe that the shift of the spectral weight to higher orders happen at the magnetic transition temperature $T_C = 45 \text{ K}$, suggesting an enhancement of electron-phonon coupling upon the development of the magnetic order.

In the paragraph starting from line 149 we discuss the observation and interpretation of this spectral weight shift of the periodic oscillations.

5. I wonder why the authors chose not to cite their own recent work [PRX 10, 011075 (2020)] and another Raman work [Nat. Nano. 15, 212 (2020)] on the same material.

We thank Reviewer #2 for noticing our recent work on CrI₃. We did not cite at the first place because we initially felt that our work (PRX 10, 011075 (2020)) and Xu group work (Nat. Nano. 15, 212 (2020)) focus on low-energy collective excitations (ours being magnon and phonon and Xu's being magnetic phonons), which is quite different from the exciton-polaron of this current manuscript.

After seeing Reviewer #2's question and rethinking in a broader context, we now think that we should have included previous Raman works done on CrI₃ to provide a comprehensive and complete picture on state-of-the-art understanding on CrI₃ by Raman spectroscopy.

In the revised manuscript, we now cite both works, as well as the other two recent Raman works (Y. Zhang *et al.*, Nano Lett. 20, 729-734 (2020) and A. McCreary *et al.*, arXiv. 1910.012137 (2019)) in line 85 and 89.

In summary, we thank Reviewer #2 for the insightful questions/comments that help improve our manuscript.

REVIEWERS' COMMENTS:

Reviewer #1 (Remarks to the Author):

The authors have revised the manuscript to address the comments by the reviewers point by point. They carefully compared different physical mechanisms of harmonic approximation model multi-phonon, cascade model multi-phonon and configuration-coordinate model multi-phonon in the revised manuscript and try to confirm that the observed Raman features in CrI₃ are indicate of the polaronic signature. They further added the temperature and magnetic-field dependent Raman spectra of 3L-5L CrI₃, which shows similar Raman features to that of bilayer CrI₃. Although I am still a bit worried about the physical origin of Raman feature that they assigned to polaronic signature since the Raman spectroscopy of monolayer CrI₃ and the resonance Raman spectra excited by the lasers with energy close to the B exciton are still missing, the authors' responses and the revisions are adequate. Other researchers in the field can perform further investigations to give an in-depth understanding about the so-called polaronic features via Raman spectroscopy. At this stage, the work is suitable for publication in Nature Communications.

Reviewer #2 (Remarks to the Author):

In the revised manuscript, the authors carefully answered the questions and comments of the reviewers as best as they can. Although some of the suggested additional measurements were not carried out due to limitations in their equipment and/or weak signal, the additional data and explanations have made the work more convincing. Although the experimental work is not complete and the theoretical model less than fully convincing, I recommend that the manuscript be accepted so that the work can stand the scrutiny of the scientific community.

Point-by-point response to Reviewers' comments

Reviewer #1 (Remarks to the Author):

The authors have revised the manuscript to address the comments by the reviewers point by point. They carefully compared different physical mechanisms of harmonic approximation model multi-phonon, cascade model multi-phonon and configuration-coordinate model multi-phonon in the revised manuscript and try to confirm that the observed Raman features in CrI₃ are indicate of the polaronic signature. They further added the temperature and magnetic-field dependent Raman spectra of 3L-5L CrI₃, which shows similar Raman features to that of bilayer CrI₃. Although I am still a bit worried about the physical origin of Raman feature that they assigned to polaronic signature since the Raman spectroscopy of monolayer CrI₃ and the resonance Raman spectra excited by the lasers with energy close to the B exciton are still missing, the authors' responses and the revisions are adequate. Other researchers in the field can perform further investigations to give an in-depth understanding about the so-called polaronic features via Raman spectroscopy. At this stage, the work is suitable for publication in Nature Communications.

We thank Reviewer #1 for his/her constructive comments in the previous round of review that greatly improve our manuscript and for his/her recommendation of this manuscript for publication in Nature Communications.

Reviewer #2 (Remarks to the Author):

In the revised manuscript, the authors carefully answered the questions and comments of the reviewers as best as they can. Although some of the suggested additional measurements were not carried out due to limitations in their equipment and/or weak signal, the additional data and explanations have made the work more convincing. Although the experimental work is not complete and the theoretical model less than fully convincing, I recommend that the manuscript be accepted so that the work can stand the scrutiny of the scientific community.

We appreciate the constructive questions that Reviewer #2 provided in the last round of review and thank him/her for making our manuscript an even better one and for recommending the acceptance of our manuscript in Nature Communications.